# The Comparative Superiority of SARS-CoV-2 Antibody Response in Different Immunization Scenarios

**DOI:** 10.3390/jpm12111756

**Published:** 2022-10-23

**Authors:** Ourania S. Kotsiou, Nikolaos Karakousis, Dimitrios Papagiannis, Elena Matsiatsiou, Dimitra Avgeri, Evangelos C. Fradelos, Dimitra I. Siachpazidou, Garifallia Perlepe, Angeliki Miziou, Athanasios Kyritsis, Eudoxia Gogou, George D. Vavougios, George Kalantzis, Konstantinos I. Gourgoulianis

**Affiliations:** 1Faculty of Nursing, School of Health Sciences, University of Thessaly, GAIOPOLIS, 41110 Larissa, Greece; 2Department of Respiratory Medicine, Faculty of Medicine, School of Health Sciences, University of Thessaly, BIOPOLIS, 41110 Larissa, Greece; 3Primary Healthcare, Internal Medicine Department, Amarousion, 15125 Athens, Greece; 4Public Health & Vaccines Lab, Department of Nursing, School of Health Sciences, University of Thessaly, GAIOPOLIS, 41110 Larissa, Greece

**Keywords:** antibody, COVID-19, infection, immunization, vaccination

## Abstract

Background: Both SARS-CoV-2 infection and/or vaccination result in the production of SARS-CoV-2 antibodies. We aimed to compare the antibody titers against SARS-CoV-2 in different scenarios for antibody production. Methods: A surveillance program was conducted in the municipality of Deskati in January 2022. Antibody titers were obtained from 145 participants while parallel recording their infection and/or vaccination history. The SARS-CoV-2 IgG II Quant method (Architect, Abbott, IL, USA) was used for antibody testing. Results: Advanced age (>56 years old) was associated with higher antibody titers. No significant differences were detected in antibody titers among genders, BMI, smoking status, comorbidities, vaccine brands, and months after the last dose. Hospitalization length and re-infection were predictors of antibody titers. The individuals who were fully or partially vaccinated and were also double infected had the highest antibody levels (25,017 ± 1500 AU/mL), followed by people who were fully vaccinated (20,647 ± 500 AU/mL) or/partially (15,808 ± 1800 AU/mL) vaccinated and were infected once. People who were only vaccinated had lower levels of antibodies (9946 ± 300 AU/mL), while the lowest levels among all groups were found in individuals who had only been infected (1124 ± 200 AU/mL). Conclusions: Every hit (infection or vaccination) gives an additional boost to immunization status.

## 1. Introduction

The Coronavirus disease pandemic 2019 remains an excellent concern for ethnicities. It is already well-established that the SARS-CoV-2 virus is rapidly evolving and spreading through mutagenesis, a quite threatening condition that lengthens the duration of the pandemic and might affect the efficacy of the existing vaccines and lead to the need to develop new ones in order to confront new variants of the specific viral infection [1,2].

There is a debate regarding the durability of antibody responses over time in patients infected by SARS-CoV-2, with several studies reporting stable, long-lasting antibody immunity and others showing rapidly waning antibody immunity or late appearances with low antibody levels and/or a complete lack of antibodies [3]. 

FDA decided on booster vaccines because the benefits of the COVID-19 vaccination far outweigh the potential risks. However, further studies are needed to demonstrate the efficacy of booster vaccinations to determine the best dosing and mix-and-match schedules of vaccinations [3]. Nevertheless, the result of the combination of infection and vaccination on the antibody levels is unknown and leads to a condition of questioning and concern.

In this study, we aimed to compare the titers of antibodies against SARS-CoV-2 in different scenarios for antibody production, which is of great importance, especially in the era of the pandemic in which we possess certain preventive tools such as vaccines.

## 2. Materials and Methods

A surveillance program was conducted in the semi-closed municipality of Deskati in January 2022. To assess the different scenarios for antibody production, antibody titers were obtained from participants while recording their infection and/or vaccination history since the pandemic wave initiation in the community in October 2020.

All the residents of Deskati were invited to participate in this program by the local authority and were notified of the time and place. Participants were recruited by announcing the research in the media, while local officials organized a one-month recruitment campaign. There were no exclusion criteria. The participants were analyzed to evaluate seroprevalence and antibody-response longevity to the SARS-CoV-2 infection and/or vaccination.

All subjects provided written and oral informed consent. Following consent, demographic information and data regarding past PCR-confirmed COVID-19 infection and vaccination history were recorded on questionnaire forms for all participants. 

The SARS-CoV-2 IgG II Quant method (Architect, Abbott, IL, USA) was used for antibody testing. This is an automated two-step chemiluminescent microparticle immunoassay that was used for the qualitative and quantitative determination of IgG antibodies against the spike receptor-binding domain (RBD) of SARS-CoV-2 in the serum specimens, with a sensitivity of 99.9% and specificity of 100% for detecting the IgG antibodies generated by prior infection or vaccination, as previously described [4,5]. The sequence used for the receptor-binding domain was taken from the WH-Human 1 coronavirus, GenBank accession number MN908947. The analytical measurement interval is stated as 21 to 40,000 AU/mL, and the positivity cutoff as ≥50 AU/mL (manufacturer defined) [6].

The Pearson correlation method was used for correlation analysis between the pairs of continuous variables. Stepwise multiple linear analysis was conducted with numerical and categorical variables turned into dummy variables. It was used to analyze the correlation between antibody titers and various factors affecting the population. The mean age, gender, mean BMI, smoking status, presence of comorbidities, previous infection, hospitalization, mean length of hospitalization, re-infection, vaccination status, brand name of the vaccine, number of vaccination doses, and months after the last vaccine dose were used as independent variables in the prediction of antibody titers. To identify differences the between two independent groups, an unpaired *t*-test was used. Parametric data comparing three or more groups were analyzed with a one-way ANOVA and Tukey’s multiple comparisons test, while non-parametric data were analyzed with the Kruskal–Wallis test and Dunn’s multiple comparison test. Pearson’s chi-squared test was used to determine whether there was a statistically significant difference between the frequencies. A result was considered statistically significant when the *p*-value was <0.05. Data were analyzed and visualized using SPSS Statistics v.23 (Armonk, NY, USA: IBM Corp.) and Tableau (Tableau Software LLC, Seattle, WA, USA), respectively.

## 3. Results

In this study, 145 participants were recruited. The main characteristics of the study population are presented in Table 1. As shown, females had more comorbidities than males. None of the participants were immunocompromised. Half of the population had previously been infected by SARS-CoV-2 for one year. A total amount of 8.1% of the infected population (*n* = 12) had a recent double infection (in the last three months) during that year, from which ten were fully vaccinated while two were not vaccinated at all. 

Most of the population (93.1%, *n* = 135) were vaccinated. A total of 82.2% (*n* = 111) were fully vaccinated (with three doses), and the rest were partly vaccinated. A total of 70.3% (*n* = 95) of the population has been vaccinated with Pfizer/BioNTech, 29% (*n* = 39) by Moderna and 0.7% by Johnson & Johnson, with no difference between genders. We found no difference in antibody titers between the different brands of vaccines (Pfizer/BioNTech vs. Moderna, 14,644 ± 11,567 vs. 10,793 ± 11,596, *p* = 0.084). There was no correlation between the months after the last vaccine dose and antibody titers (r = −0.027, *p* = 0.761). 

No difference in antibody production was observed among the genders after 27 months (*p* = 0.193). No correlation was found between BMI and antibody titers (r = 0.92, *p* = 0.293). No significant differences were detected in antibody titers by tobacco use (current and ex-smokers vs. nonsmokers, *p* = 0.522) and comorbidities (*p* = 0.073). Advanced age (>56 years old) was associated with higher antibody titers compared to younger adults (14,595 ± 12,869 AU/mL vs. 10,517 ± 9735 AU/mL, *p* = 0.039). 

SARS-CoV-2 seropositivity was 93.1% in the study population. Seronegative (*n* = 10) were only infected but unvaccinated. More specifically, the infected and not vaccinated people had no seropositivity one year after the SARS-CoV-2 infection. The winners in antibody production were the patients who were fully or partly vaccinated and had also been infected twice (Table 2), followed by people who were fully vaccinated or partially vaccinated and were infected once, with no significant difference between the last two groups. People who were vaccinated had lower antibody levels, while individuals who had only been infected had the lowest antibody titers. A statistical analysis of the different immunization scenarios revealed a significant difference in antibody titers between the groups.

Stepwise multiple linear analysis was used to analyze the correlation between antibody titers and the various factors affecting the population (Table 3). The mean age, gender, mean BMI, smoking status, presence of comorbidities, previous infection, hospitalization, length of hospitalization, re-infection, vaccination status, the brand name of the vaccine, number of vaccination doses, and months after the last vaccine dose were used as independent variables in the prediction of antibody titers. The hospitalization period and re-infection were independent predictor variables of antibody titers, explaining 52% of the total variance in this regression model. There was no multicollinearity between the explanatory variables.

## 4. Discussion

In this study, for the first time, we investigated the different scenarios for antibody production among immunocompetent participants by recording their infection and/or vaccination history during a one-year period. In particular, we found that advanced age (>56 years old) was associated with higher antibody titers. No significant differences were detected in antibody titers among genders, sex, BMI, smoking status, comorbidities, vaccine brands, and months after the last shot. Hospitalization periods and re-infection were independent predictor variables of antibody titers. Individuals who were fully or partially vaccinated and were also double infected had the highest antibody levels, followed by people who were fully vaccinated or/partially vaccinated and were infected once. People who were only vaccinated had lower levels of antibodies, while the lowest levels among all the groups were found in individuals who had only been infected. A significant difference was detected between all the groups.

Interestingly, SARS-CoV-2 seropositivity was 93% in the study population. Seronegative people were only infected with the virus and remained unvaccinated. The longevity of the antibody response to the SARS-CoV-2 infection are not well defined. We have recently reported that antibody responses to the SARS-CoV-2 infection were maintained nine months after the pandemic and especially in those with severe disease leading to hospitalization [4,5]. A recent study identified the over one year duration of SARS-CoV-2 antibodies in 82.90% of 538 convalescent COVID-19 patients [7,8]. Similarly, other studies supported a long-lasting immunological memory against SARS-CoV-2 one year after mild COVID-19 [9,10]. Conversely, in this study all the people who were infected (*n* = 11) but not vaccinated had no seropositivity after one year [9]. One large study showed that 13% of individuals lost detectable IgG titers 10 months post-infection. Yan et al. documented that SARS-CoV-2-specific IgG persistence and titer depended on COVID-19 severity, as 74.4% of recovered asymptomatic carriers had negative anti-SARS-CoV-2 IgG test results, while many others had very low virus-specific IgG antibody titers, among a population of 473 previously infected patients [11]. Hence, further studies are needed to clarify this field.

Multiple vaccine constructs have been quite promising, with an approximately 95% protective efficacy against COVID-19 [12]. Since identifying the Omicron variant, many countries have made modifications to their vaccination programs by including the recommendation of a third and fourth injection of boosting vaccination dosages in large populations to reduce the risk of adverse effects. However, all three vaccine producers (Johnson et Johnson, BioNTech, Pfizer, and Moderna) have published statements claiming vaccines would protect against severe sickness, as well as the fact that variant-specific vaccinations and boosters are in the works [13]. Nevertheless, it is unknown how long the immunity following COVID-19 vaccinations last, and this is a quite provoking situation that leads to an increased feeling of uncertainty and disbelief. It has been supported that the antibody persistence time of the mRNA vaccine is about 180 days (six months), following the adenovirus vaccine with 90 days [14]. Moreover, a short duration of antibody persistence of about 2-months has been found after the second dose [14]. 

Roy et al. reported significant differences in neutralizing antibody titers after 180 days in age, sex, COVID-19 infection, tobacco use, and asthma patients [15]. Swartz et al. measured antibody titers in 4553 participants over 11 months and documented that individuals may remain antibody positive from natural infection beyond 500 days, depending on age and smoking or vaping use [15]. Conversely, in our study, no significant differences were detected in antibody titers by sex, BMI, smoking status, and comorbidities. No significant differences were also detected in antibody titers among different brands of vaccines and months after the last shot. There are supporting data that prove there is a significantly higher humoral immunogenicity of the SARS-CoV-2 mRNA-1273 vaccine (Moderna) compared with the BNT162b2 vaccine (Pfizer-BioNTech) in infected as well as uninfected participants and across age categories [16].

An interesting finding was that advanced age (>56 years old) was associated with higher antibody titers. However, age was not a predictor of antibody titers in the stepwise multiple linear analysis. Yang et al. investigated the antibody test results among 31,426 patients from a wide range of age groups and supported an age-dependent variation in antibody titers, with children having higher antibody-binding avidity compared with young adults, but the difference was not significant [17]. However, contradictory data also exist [18,19]. Although there is an expectation that COVID-19 will become endemic, the pandemic will not end with the virus disappearing, and many questions remain unanswered. Further studies are needed to clarify the arising issues.

Moreover, in this study, the hospitalization period and re-infection were independent predictor variables of antibody titers. Similarly, Klein et al. found that hospitalization for severe COVID-19 could predict greater antibody responses against SARS-CoV-2. [18]. At present, it is unclear how long serum antibodies persist after reinfection [20]. Townsend et al. supported the fact that reinfection by SARS-CoV-2 under endemic conditions would likely occur between 3months and 5.1 years after the peak antibody response, with a median of 16 months [20]. However, to identify the correlates of protection, the relationship between in vitro neutralization levels of anti-SARS-CoV-2 antibodies and protection from severe acute respiratory syndrome coronavirus 2 (SARS-CoV-2) infection by large convalescent cohorts should be tested. Khoury et al. documented that neutralizing antibody levels are highly predictive of immune protection from symptomatic SARS-CoV-2 infection and estimated the neutralization level for 50% protection against detectable SARS-CoV-2 infection to be 20.2% of the mean convalescent level, predicting that over the first 250 days after immunization a significant loss in protection from SARS-CoV-2 infection will occur, although protection from severe disease should be largely retained [21]. However, how high a titer is protective of further infection remains unclear, and we cannot provide conclusive evidence that these antibody responses protect from reinfection. However, we believe it is very likely that higher titers will decrease the odds ratio of reinfection and may attenuate disease in the case of breakthrough infection. Undoubtedly, it is imperative to swiftly perform studies to investigate and establish a correlate of protection from SARS-CoV-2 infection.

Higher antibody titers were found in cases of vaccination in previously infected subjects, according to a previous study by our scientific team [4,5]. This finding is also supported by many other studies which report that a low concentration of SARS-CoV-2 spike protein antibodies after 9–12 months indicates that re-exposure to the virus or vaccination is required to use the B-cell immunity to full capacity [22]. In the current study, we found a 15 times higher titer of anti-SARS-CoV-2 antibodies in individuals who were fully or partially vaccinated and who were double infected than previously infected patients, while fully vaccinated patients who were infected once had 25 times higher titers of anti-SARS-CoV-2 antibodies than previously infected patients. Interestingly, patients who were only vaccinated had nine times greater antibody titers than the only-infected patients. These results reflect those of Teresa Vietri et al., who also found that a booster dose resulted in a marked increase in antibody response, which then subsequently decreased over time [23]. 

Our study’s findings should be interpreted within the context of its limitations and strengths. As such, when considering absolute numbers, our study’s population is smaller compared to other studies [20,21]. It does, however, represent a specific epidemiological framework in rural Greece, reflecting remote populations differentially affected by the pandemic. Within these parameters, albeit nested, our study reports on real-world data representative of the geographical, cultural, and healthcare settings from which they stem. Furthermore, as previously mentioned, the corroboration of our findings in larger cohorts reflects that these data may be generalizable in similar settings. Another limitation was that the sample group was rather uniformly young and very lean, which limits the generalizability of our findings. It would be appealing to apply this search among individuals of a more significant number and in different geographical areas. This could give us the unique opportunity to evaluate and more profoundly assess the potential fluctuation between the titers of antibodies against SARS-CoV-2 in different scenarios in terms of antibody production and environmental conditions. In addition, it would be intriguing to study the titers of antibodies against SARS-CoV-2 in different statuses concerning the production of antibodies in various eating habits and lifestyles. 

Last but not least, it would be particularly thrilling if we could develop a score or index concerning the antibody titers, the viral infection’s different statuses related to the antibodies produced, and the patient’s clinical image in order to have a potential prognostic tool, especially in individuals living with many comorbidities. Using scores or indices such as these, it might be possible to detect early the need for further and more specialized medical intervention and care in subjects infected with SARS-CoV-2 and vaccinated, fully or not, against the virus which has invaded our everyday routine. This could probably be beneficial not only for the infected individuals and their families but also for the healthcare system that has sustained a tremendous burden, both economically concerning every country worldwide and psychologically, especially for healthcare workers, due to the pandemic. We seem to have a long road to cross for understanding and decoding the mechanisms concerning this viral infection and its effect on human body systems.

## 5. Conclusions

The winners in anti-SARS-CoV-2 titers were the individuals who were fully or partially vaccinated and who were also double infected, followed by people who were fully vaccinated or/partially vaccinated and only infected once. In addition, subjects who were only vaccinated had lower levels of antibodies, whilst the lowest levels among all the groups were found in individuals who had only been infected. Overall, these results are quite promising, but the SARS-CoV-2 variant seems to be the dragon in this medical issue.

Our findings indicate that every hit (infection or vaccination) gives an additional boost in immunization status. However, the antibody response raised by vaccines is roughly affected by not only the time but also the emergence of new SARS-CoV-2 variants. The spread of new variants is associated with an escape from antibodies; therefore, to mitigate the spread of this infection in the long run, a more effective longitudinal observation of the immune response is needed.

## Figures and Tables

**Table 1 jpm-12-01756-t001:** Characteristics of the study population stratified by gender (N = 145).

Variable	Total (N = 145)	Males (*n* = 58)	Females (*n* = 87)	*p*-Value
Age (years)	56.0 ± 14.7	59.0 ± 16.0	54 ± 14	0.082
BMI (mg/kg^2^)	19.0 ± 2.0	19.7 ± 2.6	19.3 ± 3.5	0.567
Comorbidities yes, *n* (%)	68 (46.9)	21 (36.2)	47 (54.0)	0.041
Medication yes, *n* (%)	74 (51.0)	24 (41.4)	50 (57.5)	0.065
Previous infection yes, *n* (%)	73 (50.0)	30 (51.7)	43 (49.4)	0.382
Vaccination yes, *n* (%)	135 (93.1)	55 (94.8)	80 (91.9)	0.364
Seropositivity yes, *n* (%)	135 (93.1)	55 (94.8)	80 (91.9)	0.364
Antibody titers (AU/mL)	12,663 ± 11,725	14,229 ± 11,996	11,654 ± 11,522	0.193

**Table 2 jpm-12-01756-t002:** Antibody response in different immunization scenarios during a year period in immunocompetent population (N = 145) and statistical analysis of different immunization scenarios.

Immunization Scenario	N = 145	Titers of Anti-SARS-CoV-2 Antibodies (AU/mL)	*p*-Value *	*p*-Value **	*p*-Value ***	*p*-Value #
Fully or partially vaccinated and double infected	11	25,017 ± 1500		0.023	0.015	0.012
Fully vaccinated and infected once	44	20,647 ± 500	0.023		0.042	0.004
Partially vaccinated and infected once	8	15,808 ± 1800	0.015	0.042		0.025
Only vaccinated	71	9946 ± 300	0.012	0.004	0.025	
Only infected	11	1124 ± 200	<0.001	<0.001	<0.001	0.014

Note: * One-way ANOVA compares the means of the antibody titers of fully or partially vaccinated and double infected people with all the other independent immunization scenarios; ** One-way ANOVA compares the means of the antibody titers of fully or partially vaccinated and once infected people with all the other independent immunization scenarios; *** One-way ANOVA compares the means of the antibody titers of partially vaccinated and once infected people with all the other independent immunization scenarios; # One-way ANOVA compares the means of the antibody titers of only vaccinated people with all the other independent immunization scenarios.

**Table 3 jpm-12-01756-t003:** Stepwise multiple linear analysis between antibody titers and significant predictors.

Coefficients ^a^					
Model	Unstandardized Coefficients		Standardized Coefficients	t	*p*-Value
	B	Std. Error	Beta		
(Constant) Re-infection (yes) Length of hospitalization (days)	−4905.6	5925.1		−0.828	0.413
	13,719.1	3768.7	0.507	3640	0.001
	290.5	132.5	0.305	2192	0.034

^a^ Dependent variable: antibody titers (AU/mL), R = 52.2%, R^2^ = 27%, R^2^ (adjusted) = 23%.

## Data Availability

The data that support the findings of this study are available on request from the corresponding author, O.S.K.

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
