# Peer review of "The Comparative Superiority of SARS-CoV-2 Antibody Response in Different Immunization Scenarios"

_jpm, 2022, doi:10.3390/jpm12111756_

Round 1

Reviewer 1 Report

This paper focuses on a comparative analysis of the immune response when different vaccination regimens are used. The work is of interest for further research in personalized medicine.

Unfortunately, in this case the reviewer cannot be fully objective as there are no mathematical methods available. However, it should be noted that the paper presents and interprets the results of the experiment, which can be used for the treatment and prevention of Covid-19.

Comments can include:

1. It would be interesting to conduct a similar experiment on a larger sample size. As in this case, statistical processing of the data is possible, which would give representative results.

2. It would be interesting to see a further experiment that includes an analysis of the various factors affecting the population in which the experiment was conducted.

3. I would like to see a description of the method used in the paper. Right now there is only a reference to it.

  4. It would also be interesting to analyze how the type/brand of vaccine affects the immune response.

Author Response

Response to Reviewer 1 Comments:

  1. This paper focuses on a comparative analysis of the immune response when different vaccination regimens are used. The work is of interest for further research in personalized medicine. Unfortunately, in this case the reviewer cannot be fully objective as there are no mathematical methods available. However, it should be noted that the paper presents and interprets the results of the experiment, which can be used for the treatment and prevention of Covid-19.

RESPONSE: We sincerely thank you for your kind words about our paper. We are delighted to receive positive feedback from you.

  1. Comments can include: It would be interesting to conduct a similar experiment on a larger sample size. As in this case, statistical processing of the data is possible, which would give representative results.

RESPONSE: Thank you for this comment. We agree that it would be interesting to conduct a similar experiment on a larger sample size. Our team plans to conduct similar research in a larger population, as reported in page 6, lines 267-282.

  1. It would be interesting to see a further experiment that includes an analysis of the various factors affecting the population in which the experiment was conducted.

RESPONSE: Thank you for the comment. Αn analysis of the various factors recorded in this study has been presented in the revised manuscript, as suggested. No difference in antibody production was observed among genders. No significant differences were detected in antibody titers by tobacco use, BMI, the presence of comorbidities, and the vaccine brand. There was no correlation between months after the last vaccine dose and antibody titers. However, our analysis showed that advanced age (>56 years old) was associated with higher antibody titers, as reported on page 3, lines 114-126. Furthermore, a stepwise multiple linear analysis was used to analyze the correlation between antibody titers and various factors affecting the population (Table 3). The mean age, gender, mean BMI, smoking status, presence of comorbidities, previous infection, hospitalization, hospitalization period, re-infection, vaccination status, the brand name of the vaccine, the number of vaccination doses, the time after the last vaccine dose were used as independent variables in the prediction of antibody titers. Only hospitalization period and re-infection were independent predictors of antibody titers, explaining 52% of the total variance in this regression model. There was no multi-collinearity between the explanatory variables. This analysis is presented on page 5, lines 156-164, and Table 3 in the revised manuscript. As we have already mentioned, our team plans to conduct similar research in a larger population as we totally agree that it would be interesting to see a further experiment that includes an analysis of the various factors affecting a larger population.

  1. I would like to see a description of the method used in the paper. Right now there is only a reference to it.

RESPONSE: Thank you for the remark. In the revision we have added more information regarding the method (page 2, lines 82-85), as suggested.

  1. It would also be interesting to analyze how the type/brand of vaccine affects the immune response.

RESPONSE: Thank you fir this great suggestion. In the revision, we have added this information (page 3, lines 114-118).

We appreciate you taking the time to offer us your insights related to the paper. We found your feedback very constructive. We tried to be responsive to your concerns.

Reviewer 2 Report

This manuscript, by Ourania S. Kotsiou, et al., reported the comparative titers of SARS-CoV-2 antibody in different vaccination/infection scenarios with 145 participants in a semi-closed community in Greece for a period of 15 months.  The description of study population and the antibody titers detected among different subgroups are summarized in clearly in Table 1 and 2.  The conclusion of this manuscript is supported by the data.   Although the sample size is small, the clinical report and its indication could benefit this field of research. 

My main critiques are related to the discussion section, some added revision could strengthen this manuscript and make this result more clinically relevant to other communities. 

·       According to Table 1, this sample group is rather uniformly young and very lean with no differences between sexes.  This point should be added to discussion. 

·       About double infection, are there data available with this group the time (months) in between the 2 infections, and do you have a way to predict or known levels of the antibody titers before the 2nd infection? 

·       According to a widely cited article https://doi.org/10.1038/s41591-021-01377-8  published on Nature Medicine in 2021, neutralizing antibody levels are highly predictive of immune protection, how high of a titer in this cohort are protective of a further infection? 

All these points should be attempted to be addressed and be added to the manuscript.

Author Response

Response to Reviewer 2 Comments:

  1. This manuscript, by Ourania S. Kotsiou, et al., reported the comparative titers of SARS-CoV-2 antibody in different vaccination/infection scenarios with 145 participants in a semi-closed community in Greece for a period of 15 months. The description of study population and the antibody titers detected among different subgroups are summarized in clearly in Table 1 and 2.  The conclusion of this manuscript is supported by the data.   Although the sample size is small, the clinical report and its indication could benefit this field of research.

RESPONSE: We sincerely thank you for your kind words about our paper. We are delighted to receive positive feedback from you.

  1. My main critiques are related to the discussion section, some added revision could strengthen this manuscript and make this result more clinically relevant to other communities.

RESPONSE: Τhank you for this comment. In the revision, we have tried to strengthen the discussion section, as suggested. We hope the manuscript after careful revisions meet your standards.

  1. According to Table 1, this sample group is rather uniformly young and very lean with no differences between sexes. This point should be added to discussion.

RESPONSE: Thank you for the comment. In the revised manuscript, we have discussed the points you raised. Furthermore, an interesting point was added after a new analysis; that advanced age (>56 years old) was associated with higher antibody titers.

  1. About double infection, are there data available with this group the time (months) in between the 2 infections, and do you have a way to predict or known levels of the antibody titers before the 2nd infection?

RESPONSE: We thank you for your comment. We have available data regarding the time interval between the first and second infection, given that this was the third serosurveillance program organized in this community.

The first serosurveillance program was performed three months after the first severe pandemic wave that hardly hit the community of Deskati in November 2021 (Kotsiou OS, et al. Understanding COVID-19 Epidemiology and Implications for Control: The Experience from a Greek Semi-Closed Community. J Clin Med. 2021 Jun 23;10(13):2765).

The second program was conducted nine months after the first pandemic wave (Kotsiou OS et al. Defining Antibody Seroprevalence and Duration of Humoral Responses to SARS-CoV-2 Infection and/or Vaccination in a Greek Community. Int J Environ Res Public Health. 2021 Dec 31;19(1):407).

The present program was conducted 27 months after the first pandemic wave.

Although there are data regarding the antibody titers at each time point, the sample size of those who have been reinfected is small (only 14 participants) to be helpful in providing reliable results. In Table 1, we present these data.

Antibody titers after re-infection (AU/mL)

Antibody titers after the first infection in November 2021 (AU/mL)

Time of re-infection since the first infection (months)

22346,1

1391,3

14

31514

653,6

20

40000

886,9

23

22316,5

13142,6

12

19614,6

1000

10

23482,4

12000,4

25

40000

20000

20

24559,7

1850,8

21

2871,1

248

15

40000

 21673,4

22

40000

12380,2 

19

3000

654

12

32678,5

3869,9

24

8578,1

1200,9

18

In fact, these measurements would only be possible in the setting of a clinical trial in a larger population.

  1. According to a widely cited article https://doi.org/10.1038/s41591-021-01377-8  published on Nature Medicine in 2021, neutralizing antibody levels are highly predictive of immune protection, how high of a titer in this cohort are protective of a further infection?

RESPONSE: Thank you for this comment. We discuss this issue on the revised manuscript, as suggested (page 6, lines 237-248).

  1. All these points should be attempted to be addressed and be added to the manuscript.

RESPONSE: We appreciate all of your insightful comments. We found them quite useful as we approached our revision. We are grateful for the time and energy you expended on our behalf.

Reviewer 3 Report

The authors of this manuscript used the Abbott SARS-CoV-2 IgG II Quant assay to measure the antibody titer against SARS-CoV-2 in 145 participants in Deskati in January 2022, attempting to compare the antibodies levels among people in different scenarios. The results showed that, while SARS-CoV-2 infection only induced a low level of antibodies, significantly lower than vaccination, it can boost the titer of antibodies in vaccinated individuals. This is consistent with previous reports.

Although this study was carefully carried out, it does have some limitations.

1. The number of participants is small and insufficient to draw a firm conclusion.

2. There is no time course of vaccination or infection, which is critical to antibody titer.

3. There is no statistical analysis of different immunization scenarios.

Author Response

Response to Reviewer 3 Comments:

  1. The authors of this manuscript used the Abbott SARS-CoV-2 IgG II Quant assay to measure the antibody titer against SARS-CoV-2 in 145 participants in Deskati in January 2022, attempting to compare the antibodies levels among people in different scenarios. The results showed that, while SARS-CoV-2 infection only induced a low level of antibodies, significantly lower than vaccination, it can boost the titer of antibodies in vaccinated individuals. This is consistent with previous reports. Although this study was carefully carried out, it does have some limitations. The number of participants is small and insufficient to draw a firm conclusion.

RESPONSE: We thank you for your comment. Our study indeed represents a specific cultural, geographic and epidemiological contextual framework. As such, it may not reach absolute numbers of more densely populated (i.e. urban) areas. It does however represent the collection and analysis of real-world data from its corresponding setting, with significant corroboration from other studies – which may report on more sizeable populations. Based on your suggestion, we have added both this limitation in the revised manuscript. We appreciate the both the suggestion and the opportunity to present the context of our findings.

  1. There is no time course of vaccination or infection, which is critical to antibody titer.

RESPONSE:

We thank you for your comment. Although this was a study based on real-world data rather than a clinical trial, we have available data regarding the time interval between the first and second infection, given that this was the third serosurveillance program organized in this community. The first serosurveillance program was performed three months after the first severe pandemic wave that hardly hit the community of Deskati in November 2021 (Kotsiou OS, et al. Understanding COVID-19 Epidemiology and Implications for Control: The Experience from a Greek Semi-Closed Community. J Clin Med. 2021 Jun 23;10(13):2765).

The second program was conducted nine months after the first pandemic wave (Kotsiou OS et al. Defining Antibody Seroprevalence and Duration of Humoral Responses to SARS-CoV-2 Infection and/or Vaccination in a Greek Community. Int J Environ Res Public Health. 2021 Dec 31;19(1):407).

The present program was conducted 27 months after the first pandemic wave.

Although there are data regarding the antibody titers at each time point, the sample size of those who have been reinfected is small (only 14 participants) to be helpful in providing reliable results. In Table 1, we present these data.

Antibody titers after re-infection (AU/mL)

Antibody titers after the first infection in November 2021 (AU/mL)

Time of re-infection since the first infection (months)

22346,1

1391,3

14

31514

653,6

20

40000

886,9

23

22316,5

13142,6

12

19614,6

1000

10

23482,4

12000,4

25

40000

20000

20

24559,7

1850,8

21

2871,1

248

15

40000

 21673,4

22

40000

12380,2 

19

3000

654

12

32678,5

3869,9

24

8578,1

1200,9

18

Furthermore, we have data regarding the time course of vaccination, a variable used for further analysis. We found no correlation between months after the last vaccine dose and antibody titers (r= -0.027, p=0.761). These data were also used as an independent variable in stepwise multiple linear analysis to analyze the correlation between antibody titers and various factors affecting the population. We appreciate both the suggestion and the opportunity to present the context of our findings.

  1. There is no statistical analysis of different immunization scenarios.

RESPONSE: Thank you for the comment. In the revised manuscript the statistical analysis for the different immunization scenarios was added, as presented in Table 2.

RESPONSE: We appreciate all of your insightful comments. We found them quite useful as we approached our revision. We are grateful for the time and energy you expended on our behalf.